

# Reporting inconsistency between published conference abstracts and article abstracts of randomised controlled trials in prosthodontics presented at IADR general sessions

Guanru Wang[1,2,*], Junsheng Chen[1,*], Honglin Li[1,2], Cheng Miao[1,2], Yubin Cao[1,3] and Chunjie Li[1,2]

[1] West China Hospital of Stomatology, Sichuan University, State Key Laboratory of Oral Diseases & National Clinical Research Center for Oral Diseases, Chengdu, Sichuan, China

[2] West China Hospital of Stomatology, Sichuan University, Department of Head and Neck Oncology, Chengdu, Sichuan, China

[3] West China Hospital of Stomatology, Sichuan University, Department of Oral and Maxillofacial Surgery, Chengdu, Sichuan, China

[*] These authors contributed equally to this work.

Corresponding authors
Yubin Cao, yubin.cao@scu.edu.cn
Chunjie Li, lichunjie@scu.edu.cn

## ABSTRACT

**Background**. There is commonly a discrepancy between conference abstracts and published article abstracts in prosthodontic randomized controlled trials (RCTs), which may mislead the scholars those attend conferences.

**Objective**. To identify the characteristics predicting inconsistency between conference abstracts and published article abstracts in prosthodontic RCTs.

**Methods**. The conference abstracts of prosthodontic RCTs presented at the IADR general sessions from 2002 to 2015 were searched. Electronic searches of MEDLINE, EMBASE, the Cochrane Library, and Google Scholar databases were conducted to match full-text publications for conference abstracts. Two investigators extracted basic characteristics and assessed the consistency and reporting quality independently and in duplicate. The linear regression model was used to analyze the predictors of inconsistency.

**Results**. A total of 147 conference abstracts were matched with published articles. Results for the secondary outcome measure, Statistical analysis, and precision measure were less than 50% consistent, and even nearly 5% of the studies had opposite conclusions. Multiple linear regression analysis showed that three factors were correlated with lower inconsistency, including continent of origin ($p = 0.011$), presentation type ($p = 0.017$), and difference in reporting quality ($p = 0.013$).

**Conclusion**. Conference attendees should cautiously treat the findings of the conference abstracts. Researchers should improve the precision of the information delivered at conferences. We recommend the authors of RCTs to explain the primary difference between conference abstracts and article abstracts.

## INTRODUCTION

Academic conferences are important for scholars to share scientific research achievements and research methods. The International Association for Dental Research (IADR) is an international dental academic organization, which was founded in 1920. With more than 11,000 memberships worldwide, IADR has been at the forefront of advancing research for the prevention of oral diseases and its academic conferences have become an important occasion for dental researchers to share basic, clinical and translational research (*Whelton & Fox, 2015*). During the conference, scientists from all over the world will present their researches to conference attendees in the form of abstracts. However, a survey showed that the full-text publication proportion of dental conference abstracts is only 29.6% (*Hua et al., 2016*). The reasons for the unpublished abstracts may be a lack of time to continue the study, the research still ongoing, etc (*Sprague et al., 2003*; *Ha et al., 2008*; *Scherer et al., 2015*). At the same time, some scholars have found that the published articles are not completely consistent to the abstracts presented at the conference (*Chalmers, Frank & Reitman, 1990*; *Van den Bogert et al., 2017*). *Wu et al. (2020)* found at least one discrepancy between the conference abstracts of European Association for Osseointegration and the published article abstracts in terms of title, statistical method, main results, and sample size. Therefore, the scientific validity and accuracy of the conference abstracts are controversial.

Randomised-controlled trials (RCT) are the gold standard in the field of evidence-based medicine (*Clancy, 2002*; *Haynes, Devereaux & Guyatt, 2002*; *Pihlstrom et al., 2012*) and the highest level of the Oxford evidence classification system (*Luksanapruksa & Millhouse, 2016*). RCTs play an important role in guiding the clinical practice. It can help doctors to make the best choice in terms of indications, diagnostic criteria, and treatment methods for specific patients (*Brignardello-Petersen et al., 2014*). However, many RCTs have unreasonable designs, improper statistical analysis, and incomplete descriptions of results (*Hua et al., 2019*; *Qin et al., 2021*). Some authors of RCTs may even spin results and distort findings (*Boutron et al., 2010*; *Guo et al., 2021*), which reduces the quality and evidence level of RCTs.

There are many RCTs in the conference abstracts (*Scherer, Langenberg & Von Elm, 2007*; *Scherer & Saldanha, 2019*). Nevertheless, conference abstracts have not undergone a prepublication peer-review process (*Schmucker et al., 2017*), so it is questionable whether the findings of conference RCTs can be used to guide clinical practice. The inconsistency of conference abstracts before and after publication also reduces the authenticity and reliability of RCTs presented at conferences. How participants judge and identify reliable conference RCTs is an issue that needs to be addressed. Prosthodontics is an important branch of dental medicine. Our previous study discovered that the full-text published proportion of the abstracts of prosthodontics RCTs presented at the IADR general sessions was only 43.24% (*Chen et al., 2020*), and the discrepancies and related risk factors between published conference abstracts and article abstracts of them have never been investigated.

Therefore, the purpose of this study are as follows: (a) to investigate the discrepancies between published conference abstracts and article abstracts of prosthodontics RCTs

presented at the IADR general sessions; (b) to explore the risk factors related to their inconsistency.

## MATERIALS & METHODS

### Selection of conference abstracts

RCT abstracts that were presented at the IADR General Sessions (2002–2015) were obtained directly from the official website (https://iadr.abstractarchives.com/home). After removing duplicate abstracts from different databases through Endnote (version X9; Thomson Corporation, Stamford, CT, USA), we screened the rest of the abstracts and included abstracts of the RCTs on therapeutic interventions that took place in the clinical context of prosthodontics, which targeted people. The exclusion criteria are *in-vitro* studies or not conducted on human, related to other specialities, pilot/feasibility studies, trial protocols, non-RCT research, follow-up studies from previous trials. In order to eliminate the impact of time on the full-text publication, avoiding bias caused by time, we set the deadline for the publication of the article as December 31, 2020.

### Retrieval of the full text of matched articles

The two investigators (G.W. and J.C.) independently and in duplicate searched the following databases: MEDLINE (*via* PubMed), EMBASE (*via* OVID), Cochrane Library, and Google Scholar. There are no language restrictions on retrieval content. Before the formal retrieval, the consistency of the two investigators was determined by the pilot study: thirty conference abstracts that met the inclusion and exclusion criteria were randomly selected by online randomization software (https://www.randomizer.org/), and then two investigators searched independently and synchronously. The consistency of the two investigators was evaluated by Cohen's $\kappa$ statistic and the overall $\kappa$ statistic was 0.93, indicating excellent agreement between them.

Full-text publications were identified as previously described in our another article (*Chen et al., 2020*). The identification of publication began with a individual search of authors' names. When the single author corresponded to multiple publications, authors' names were combined with keywords in the abstract for advanced search. Among the results, the conference abstracts and the corresponding articles that had at least one author in common were initial included. Then the study hypothesis, intervention, and conclusion between them were further screened. If the conference abstracts and corresponding articles contained substantial similarities. This abstract was classified as 'published'. The publications with dates that were the closest to the conference were included for further study. The conference abstract was considered 'unpublished' when there was no corresponding articles after searching the databases. When the views of the two investigators were controversial, a third researcher (Y.C.) was introduced to discuss and determine the results.

### Data extraction

Two investigators (G.W. and J.C.) independently and synchronously extracted data from retrieved published conference abstracts that met the criteria and counted the results

in the excel table. The extracted data include date of presentation, continent of origin, presentation type (oral *vs* poster), number of authors, sample size, exact *p* value (yes or no), center (single-center *vs* multicenter), type of institution (Universities or Other institutions), number of affiliations, overall conclusion (positive, negative, neutral), and subspecialty focus. The consolidated standards of reporting trials for abstracts (CONSORT-A) (*Hopewell et al., 2008a*; *Hopewell et al., 2008b*) was scored for both conference abstracts and article abstracts. Each reported item was scored as one and the total score was calculated.

## Evaluation of discrepancies

We investigated the discrepancies between conference abstracts and article abstracts, quantified the inconsistency between them into 12 items in total, and some items had sub-items under them. The discrepancies were evaluated independently and in duplicate by the two investigators (G.W. and J.C.). The evaluated items include title, first author, study objective, intervention, study duration, sample size, primary outcome, results for the primary outcome measure, results for the secondary outcome measure, statistical analysis, precision measure, and conclusion. The abstract was judged for each item. If the item of the conference abstract was consistent with that of the article abstract, the value was assigned to 1, and if it was inconsistent or could not be identified, the value was assigned to 0. The scores of the two were counted and calculated to obtain a gross score (0-12). In the event of controversies, the final results were discussed with the third investigator (Y.C.).

## Data analysis

Demographic characteristics of published conference abstracts were first presented. After that, the relationship between the inconsistency of abstracts and risk factors was analyzed by multiple linear regression analysis. The conference abstracts and article abstracts with the same research content were matched, and the reporting quality of the abstracts was compared by the paired *t*-test. Statistical analyses were conducted with STATA (Version 14.0, StataCorp, College Station, TX, USA).

## RESULTS

A total of 10,268 conference abstracts of IADR (2002–2015) were searched, the duplicated 6619 were removed, and 340 abstracts met the inclusion and exclusion criteria after screening the rest 3649 abstracts. Through the retrieval of the databases, 147 abstracts were later published as journal articles (Fig. 1).

Of the 147 published conference abstracts, 18 (12.24%) were presented in 2012, followed by 16 (10.88%) and 14 (9.52%) in 2010 and 2015, respectively, and only 4 (2.72%), in 2004 and 2006. Geographically, 54 (36.73%) of the published conference abstracts have been from Europe, accounting for the largest proportion, followed by North and South America, with 35 (23.81%), while Asia, Africa, and Australia have fewer published abstracts, with a cumulative total of 23 (15.65%). Poster presentations accounted for a higher proportion of published abstracts than oral presentations (57.14% *vs.* 42.86%). The mean and standard deviation (range) of authors, sample size, and number of affiliations were $5.57 \pm 2.82$ (1-21), $54.29 \pm 47.92$ (6-282), and $1.99 \pm 2.41$ (1-18) respectively. 103
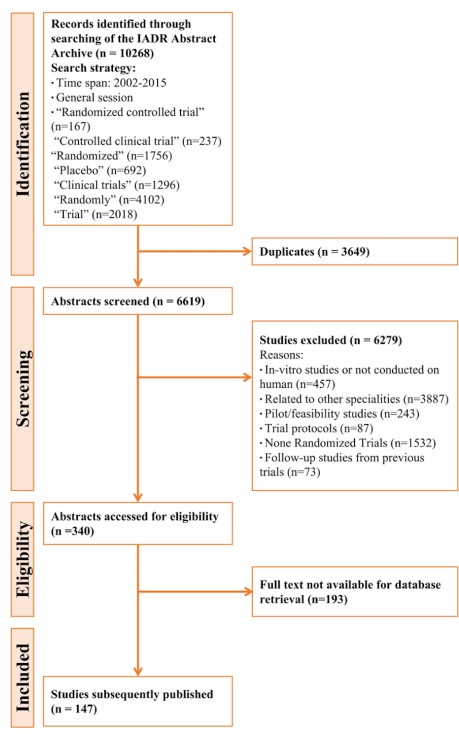

**Figure 1** Flow chart of published conference abstracts selection according to inclusion and exclusion criteria.

(70.07%) conference abstracts had the exact *p* values; 133 (90.48%) abstracts were single-center studies, and 144 (97.96%) abstracts were conducted by universities. The conclusions of 85 (57.82%) abstracts were positive, followed by neutral 44 (29.93%) and negative (12.24%). In subspecialty focus, the largest number of published conference abstracts were about complete denture and overdenture and dental composites and adhesives, both of which had 37 articles, accounting for 25.17%. The second was implant-based prosthetics and temporomandibular disorders, 24 (16.33%) and 23 (15.65%), respectively. The least subspecialty focus was removable partial dentures, with only five, accounting for 3.40% (Table 1).

Table 2 lists the discrepancies in 12 items of the 147 published abstracts. The item that was the most consistent between the conference abstracts and published abstracts was study objective (145,98.64%), followed by intervention and primary outcome, with 144 (97.96%) and 143 (9728%), respectively. In the area of precision measure, only 43 (29.25%) were identical, while 31 (21.09%) were different, and 73 (49.66%) could not be compared, as 27 (18.37%) were mentioned only in the conference abstracts, 19 (12.93%) only in the article abstracts and 27 (18.37%) in neither. Interestingly, the conclusions of 139 (95.24%) abstracts were identical, but the conclusions of seven (4.76%) abstracts were different, two (1.36%) abstracts were concluded by positive conclusions changed to negative ones, two (1.36%) abstracts were concluded by negative conclusions changed to positive ones, and even three (2.04%) abstracts were complete changed (Table 2).

**Table 1  Demographic characteristics of conference abstracts.**

| Characteristic | Category | n | n% (100%=147) |
|---|---|---|---|
| Year of presentation | 2002 IADR/AADR/CADR General Session | 11 | 7.48 |
| | 2003 IADR/PER General Session | 9 | 6.12 |
| | 2004 IADR/AADR/CADR General Session | 4 | 2.72 |
| | 2005 IADR/AADR/CADR General Session | 12 | 8.16 |
| | 2006 IADR General Session | 4 | 2.72 |
| | 2007 IADR/AADR/CADR General Session | 10 | 6.80 |
| | 2008 IADR/CADR General Session | 11 | 7.48 |
| | 2009 IADR/AADR/CADR General Session | 11 | 7.48 |
| | 2010 IADR/PER General Session | 16 | 10.88 |
| | 2011 IADR/AADR/CADR General Session | 12 | 8.16 |
| | 2012 IADR/LAR General Session | 18 | 12.24 |
| | 2013 IADR/AADR/CADR General Session | 9 | 6.12 |
| | 2014 IADR/AMER General Session | 6 | 4.08 |
| | 2015 IADR/AADR/CADR General Session | 14 | 9.52 |
| Continent of origin | Europe | 54 | 36.73 |
| | North America | 35 | 23.81 |
| | South America | 35 | 23.81 |
| | Asia/Africa/Australia | 23 | 15.65 |
| Presentation type | Oral | 63 | 42.86 |
| | Poster | 84 | 57.14 |
| Number of authors | Mean | | 5.57 |
| | Standard deviation (Range) | | 2.82 (1-21) |
| Sample size | Mean | | 54.29 |
| | Standard deviation (Range) | | 47.92 (6-282) |
| Exact p value | Yes | 103 | 70.07 |
| | No | 44 | 29.93 |
| Center | Single-center | 133 | 90.48 |
| | Multicenter | 14 | 9.52 |
| Type of institution | Universities | 144 | 97.96 |
| | Other institutions | 3 | 2.04 |
| Number of affiliations | Mean | | 1.99 |
| | Standard deviation (Range) | | 2.41 (1-18) |
| Overall conclusion | Positive | 85 | 57.82 |
| | Negative | 18 | 12.24 |
| | Neutral | 44 | 29.93 |
| Subspecialty focus | Fixed prosthodontics | 10 | 6.80 |
| | Removable partial dentures | 5 | 3.40 |
| | Complete denture and Overdenture | 37 | 25.17 |
| | Implant-based prosthetics | 24 | 16.33 |
| | Dental composites and adhesives | 37 | 25.17 |
| | Temporomandibular disorders | 23 | 15.65 |
| | Others | 11 | 7.48 |

**Table 2** Inconsistency between conference abstracts and article abstracts.

| Characteristic | Category | n (%) |
|---|---|---|
| Title | Identical | 119 (80.95) |
| | Different | 28 (19.05) |
| First author | Identical | 104 (70.75) |
| | Different | 43 (29.25) |
| Study objective | Identical | 145 (98.64) |
| | Different | 2 (1.36) |
| Intervention | Identical | 144 (97.96) |
| | Different | 3 (2.04) |
| Study duration | Identical | 95 (64.63) |
| | Different | 27 (18.37) |
| | Unable to compare | 25 (17.01) |
| | a. Only described in the conference abstract | 3 (2.04) |
| | b. Only described in the final publication | 6 (4.08) |
| | c. Not mentioned | 16 (10.88) |
| Sample size | Identical | 101 (68.71) |
| | Different | 40 (27.21) |
| | a. Increased in final publication | 35 (23.81) |
| | b. Decreased in final publication | 5 (3.40) |
| | Unable to compare | 6 (4.08) |
| | a. Only described in the conference abstract | 5 (3.40) |
| | b. Only described in the final publication | 1 (0.68) |
| Primary outcome | Identical | 143 (97.28) |
| | Different | 4 (2.72) |
| Results for the primary outcome mea- | Identical | 136 (92.52) |
| | Different | 11 (7.48) |
| Results for the secondary outcome measure | Identical | 67 (45.58) |
| | Different | 80 (54.42) |
| | a. Data added | 31 (21.09) |
| | b. Data deleted | 38 (25.85) |
| | c. Complete changed | 11 (7.48) |
| Statistical analysis | Identical | 59 (40.14) |
| | Different | 21 (14.29) |
| | Unable to compare | 67 (45.58) |
| | a. Only in the conference abstract | 30 (20.41) |
| | b. Only in the final publication | 6 (4.08) |
| | c. Not mentioned | 31 (21.09) |
| Precision measure | Identical | 43 (29.25) |
| | Different | 31 (21.09) |
| | Unable to compare | 73 (49.66) |
| | a. Only in the conference abstract | 27 (18.37) |
| | b. Only in the final publication | 19 (12.93) |
| | c. Not mentioned | 27 (18.37) |
**Table 2** (*continued*)

| Characteristic | Category | n (%) |
|---|---|---|
| | Identical | 139 (95.24) |
| | Different | 7 (4.76) |
| Conclusion | a. Positive conclusion changed to negative one | 2 (1.36) |
| | b. Negative conclusion changed to positive one | 2 (1.36) |
| | c. Complete changed | 3 (2.04) |

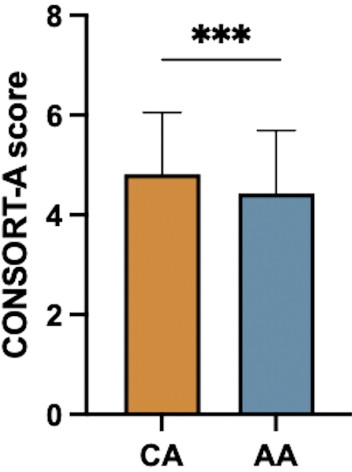

**Figure 2** **Difference of CONSORT-A score between conference abstracts and article abstracts.** Note: CA, conference abstract; AA, article abstract; ***, $p < 0.001$.

The reporting quality of conference abstracts and article abstracts was evaluated through CONSORT-A. The results of paired $t$-test showed that the mean CONSORT-A score of the conference abstracts was $4.816 \pm 1.239$, and the mean CONSORT-A score of the article abstracts was $4.429 \pm 1.266$. There was a statistical difference in the overall mean CONSORT-A score between the two groups (the difference was $-0.388$, 95% CI $\geq 0.585 \pm 0.191$, $p < 0.0002$) (Fig. 2).

The relationship between the gross score of inconsistency and risk factors was analyzed by multiple linear regression, and the interference of confounding factors is eliminated at the same time. The results showed that only three of the six independent variables were correlated with the gross score, which were continent of origin ($p = 0.011$), presentation type ($p = 0.017$), and the absolute value of CONSORT-A difference ($p = 0.013$) (Table 3).

# DISCUSSION

The ultimate criterion to evaluate the quality of a conference abstract is whether it is published in a peer-reviewed journal (*Prasad et al., 2012*; *Neves, Lavis & Ranson, 2012*). However, not all conference abstracts are later published as full-text articles (*Stranges et al., 2015*; *Chen et al., 2020*; *Hinrichs, Ramirez & Ameen, 2021*). In addition, *Yoon & Knobloch (2012)* found that compared to conference abstracts, article abstracts had at least one minor difference in title or authorship and 65% of article abstracts had major differences in study

**Table 3  Multiple linear regression of consistency related predictors.**

| Predictor | Category/unit | B | 95% CI | p value |
|---|---|---|---|---|
| Follow up times | 1 month | −0.008 | (−0.018, 0.001) | 0.079 |
| Continent of origin | South America | Baseline (reference) | | 0.011[*] |
| | North America | −0.423 | (−0.917, 0.072) | |
| | Europe | −0.757 | (−1.267, −0.246) | |
| | Asia/Africa/Australia | −0.812 | (−1.387, −0.237) | |
| Presentation type | Poster | Baseline (reference) | | 0.017[*] |
| | Oral | 0.498 | (0.090, 0.906) | |
| Number of affiliations | 1 affiliation | 0.010 | (−0.078, 0.100) | 0.819 |
| Subspecialty focus | Temporomandibular disorders | Baseline (reference) | | 0.263 |
| | Fixed prosthodontics | 0.363 | (−0.372, 1.098) | |
| | Removable prosthodontics | 0.424 | (−0.827, 1.675) | |
| | Complete denture/Overdenture | −0.093 | (−0.611, 0.424) | |
| | Implant-based prosthetics | −0.049 | (−0.613, 0.514) | |
| | Dental composites and adhesives | 0.282 | (−0.240, 0.804) | |
| | Others | −0.604 | (−1.420, 0.216) | |
| Difference of CONSORT-A score | Per unit | −0.281 | (−0.502, −0.060) | 0.013[*] |

Notes.

   B, coefficient; CI, confidence interval.
   [*]$p < 0.05$.

conclusions, statistical analysis, etc. Astonishingly, according to Theman's studies, the inconsistencies of results and/or conclusions between conference abstracts and published full-length articles were 14% (*Theman, Labow & Taghinia, 2014*). The inconsistency led conference attendees to question the authenticity of the conference abstracts. We had a similar result in the prosthodontic RCTs. The items with high consistency were study objective, intervention, primary outcome, and conclusion, which reached more than 95%.

These items were the most basic framework and components of an RCT, and there was little chance of change after the study plan was established. However, it made us suspect that whether some authors changed the primary outcome and object to reach an ideal endpoint in the publications. Moreover, though rare, the credibility of conference abstracts may be decreased if conclusions of conference abstracts are changed or even reversed in the final publications.

Then, although the sample size was also a basic element of RCT, only 68.71% of abstracts were consistent before and after publication. The changes of sample size increased the possibility of discrepancy between conference abstracts and article abstracts. *Dagi et al. (2021)* found that an increase or decrease in sample size greater than 10% increased the possibility of a discrepancy by eight-fold or 25-fold, respectively. The sample size may be increased in the final publication due to the continuation of recruitment. However, it may be difficult to explain why the sample size is decreased (*Kleweno et al., 2008*). It may be attributable to that some patients should have been excluded in the recruitment screening or that some researchers may manipulate or omit the sample size in order to obtain statistically significant and positive results. The authors should indicate whether the

sample size is changed from previously reported results and explain the reason of changes clearly in the final publication to avoid the misunderstanding of academic misconduct (*Dagi et al., 2021*).

Items such as study duration, statistical analysis, results for the secondary outcome measure, and precision measure could be timely adjusted according to the progress of the project, so there were discrepancies before and after publication. However, for the transparency of publications, we suggest the authors should report all the secondary outcomes, whatever in single or multiple articles, or in the main text or supplementary materials. All the secondary outcomes reported in the conference should at least be included in the final publication.

The risk factors related to the consistency of conference abstracts before and after publication were analyzed by multiple linear regression, and the results showed that content of origin ($p = 0.011$), presentation type ($p = 0.017$), and the difference in CONSORT-A scores ($p = 0.013$) were associated with consistency scores. The pre- and post-publication variability of conference abstracts from all other continents was less than that of South America. The inconsistency was more severe for poster-presentation abstracts than for oral-presentation abstracts. Compared to poster abstracts, oral presentation abstracts were subjected to rigorous expert review and had higher study quality and scientific priority than poster abstracts, which made higher consistency of oral presentation abstracts.

The larger difference between the CONSORT-A scores before and after publication, the greater the discrepancies of the basic framework. It indicated that some items were only reported in the conference or article abstracts. The results of the paired $t$-test showed higher reporting quality for conference abstracts than for article abstracts, yet the conclusion of Uzung et al. showed higher reporting quality for article abstracts than for conference abstracts (*Yoon & Knobloch, 2012*). We speculated that this may be attributable to the requirements of word limit and abstract structure. For example, the *Journal of Dental Research* limits 300 words for abstract while the IADR conference abstract does not. Therefore, authors are allowed to describe conference abstracts in detail according to CONSORT-A, whereas they may have to omit some items and details to meet the journal's requirements. To ensure that conference submissions accurately report their studies, we recommend authors to present their abstracts closely following CONSORT (for RCTs), preferred reporting items for systematic reviews and meta-analyses (PRISMA, for meta-analyses), along with sharing their trials registration numbers, funding sources and other important informations (*Rowhani-Farid et al., 2023*).

Despite our results, previous studies also found the discrepancy may be resulted by disagreement among co-authors on the final articles (*Sprague et al., 2003*). Besides, when the authors submit their manuscripts to the journals, they make changes based on the feedback of the editors or reviewers, which may cause discrepancies between conference abstracts and article abstracts (*Prasad et al., 2012*). The difference of conflict of interest of project funds (*Weiss & Davis, 2019*) may also make changes in items such as the first author before and after the publication. Overall, the authors should report all the results in trials and explain why the final article is different from the conference version, to promote the scientific transparency.

There are still limitations in this study. First, this study only addressed prosthodontic RCTs in IADR general sessions. It may be different to infer whether our results could be generalized to other domains or subjects. Secondly, there may be articles published in the full text that were not included in the electronic database, such as local journals, or not published within the given time frame. However, our retrieval strategy is systematic and comprehensive, which ensures the most efficiency of full-text retrieval. The Cochrane review showed that the median publishing time of the RCT study was 18 months, and the publication rate decreased significantly after 3 years (*Scherer et al., 2018*). Our retrieval time was five years apart from the deadline for publication, so most articles could be published within the period. Finally, we only compared published article abstracts and conference abstracts instead of published full-text, which may ignore some important discrepancies and their reasonable explanations in the manuscript. A further study to explore the discrepancies between the conference abstracts and published manuscript is suggested to remedy the limitation.

## CONCLUSIONS

There were multiple discrepancies between the published conference abstracts of RCTs and the article abstracts of the IADR general sessions in 2002-2015. The continent of origin, presentation type, and the CONSORT-A difference was correlated with inconsistency before and after publication. Conference attendees should cautiously treat the findings of the conference abstracts. Researchers should improve the precision of the information delivered at conferences. We recommend authors of RCTs to explain the primary difference between conference abstracts and article abstracts.

### Funding
This work was supported by the 2018 Sichuan University-Luzhou Municipal Government Strategic Cooperation Research (2018CDLZ-12). The funders had no role in study design, data collection and analysis, decision to publish, or preparation of the manuscript.

### Grant Disclosures
The following grant information was disclosed by the authors:
2018 Sichuan University-Luzhou Municipal Government Strategic Cooperation Research: 2018CDLZ-12.

### Competing Interests
The authors declare there are no competing interests.

### Author Contributions
- Guanru Wang conceived and designed the experiments, performed the experiments, analyzed the data, prepared figures and/or tables, authored or reviewed drafts of the article, and approved the final draft.

- Junsheng Chen conceived and designed the experiments, performed the experiments, analyzed the data, prepared figures and/or tables, and approved the final draft.
- Honglin Li analyzed the data, prepared figures and/or tables, and approved the final draft.
- Cheng Miao performed the experiments, prepared figures and/or tables, and approved the final draft.
- Yubin Cao conceived and designed the experiments, prepared figures and/or tables, authored or reviewed drafts of the article, and approved the final draft.
- Chunjie Li conceived and designed the experiments, prepared figures and/or tables, and approved the final draft.

## Data Availability

The raw material is available in the Supplementary Files. The raw data shows the published abstracts that were concluded in this article.

## Supplemental Information

Supplemental information for this article can be found online at http://dx.doi.org/10.7717/peerj.15303#supplemental-information.

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
