# Peer review of "Reporting inconsistency between published conference abstracts and article abstracts of randomised controlled trials in prosthodontics presented at IADR general sessions"

_PeerJ, doi:10.7717/peerj.15303_

## Round 0.1 · original submission · Major Revisions

Based on the Reviewers' comments, the article needs major revision.

·

Basic reporting

no comment

Experimental design

In the methodology section, it is not mentioned who are the two investigators who were checking the study, and who was the third investigator. In systematically conducted studies like these, it is preferred that the investigators are mentioned this; therefore, the authors may specify this

Validity of the findings

Lines 197-198: “…. It may be attributable to that some patients should have been excluded in the recruitment screening….”
I feel there is another possibility here. Some patient data may have been manipulated (or omitted) to reach desired results (statistical significance, or lack thereof). The authors can give their opinion on this

In lines 141-142, the authors mention: “……… The average number of authors, the average sample size, and 142 the average number of affiliations were 5.57 ±2.82 (1-21), 54.29 ±47.92 (6-282), and 1.99 ±2.41….”
The authors should clarify here how they are expressing these statistics. I suppose it is ‘median +- interquartile range (range)’. It is usually recommended that these should be mentioned

In the discussion section, the authors can compare their results with more similar research works done in other fields. Some such examples are:

- Conference abstract vs publication abstract: Dagi AF, Parry GJ, Labow BI, Taghinia AH. Discrepancies between Conference Abstracts and Published Manuscripts in Plastic Surgery Studies: A Retrospective Review. Plast Reconstr Surg Glob Open. 2021 Sep 17;9(9):e3828. doi: 10.1097/GOX.0000000000003828. PMID: 34549011; PMCID: PMC8448048.

- Conference abstract vs full-text abstract:
Rowhani-Farid A, Hong K, Grewal M, Reynolds J, Zhang AD, Wallach JD, Ross JS. Consistency between trials presented at conferences, their subsequent publications and press releases. BMJ Evid Based Med. 2022 Nov 10:bmjebm-2022-111989. doi: 10.1136/bmjebm-2022-111989. Epub ahead of print. PMID: 36357160.

- Prasad S, Lee DJ, Yuan JC, Barao VA, Shyamsunder N, Sukotjo C. Discrepancies between Abstracts Presented at International Association for Dental Research Annual Sessions from 2004 to 2005 and Full-Text Publication. Int J Dent. 2012;2012:859561. doi: 10.1155/2012/859561. Epub 2012 Feb 22. PMID: 22505912; PMCID: PMC3296196.

- Theman TA, Labow BI, Taghinia A. Discrepancies between meeting abstracts and subsequent full text publications in hand surgery. J Hand Surg Am. 2014 Aug;39(8):1585-90.e3. doi: 10.1016/j.jhsa.2014.04.041. Epub 2014 Jun 13. PMID: 24934603.

Reviewer 2 ·

Basic reporting

Clear, unambiguous, professional English language used throughout.
• The English language could be improved in parts to clarify your meaning and fix awkward phrasing. See the lines below:
o 40-41 and in conclusions section: Use active voice -- “We recommend that authors of RCTs …”
o 50-51: “The reasons for the unpublished abstracts…”
o 56, 63: remove “and so on.” Instead write out the reasons
o 66: “RCTs have spins” - could change to “some authors of RCTs may spin results and distort findings”
o 71: Remove “of course” which is informal English
o 134-135: Clarify this sentence. You could just say 147 abstracts were later published as journal articles.
o 151-153: Suggest changing this sentence into two sentences. “Table 2 lists…”, and then “The items that were the most consistent between the conference abstracts and published abstracts were…”
o 180-181: I think “Not all conference abstracts are available for publication” is better phrased as “Not all conference abstracts are later published as full-text articles.”
o 196: suggest replacing “weird and uneasy” with “difficult to explain why”
o 198: remove “but whatever” which is too informal
o 223-224: Sentence needs some clarification.

Intro & background to show context. Literature well referenced & relevant.
• In the introduction, the authors demonstrated the context of their research study, including how conference abstract discrepancies reduce the reliability of RCTs presented at conferences, and calls into the question whether practitioners should use the results to guide their practice. A few minor suggestions are below.
1. The purpose of the study is addressed in lines 74-76 but could use an additional sentence clearly indicating your research aims.
2. I suggest describing why you selected the IADR conference as opposed to other dental conferences. Does it have the largest attendance? Is there is a large proportion of practitioners that attend and may make use of the research presented in practice?
3. Consider also adding why you selected prosthodontics abstracts as opposed to other research areas.

Figures are relevant, high quality, well labelled & described.
Figure 1
• This flow diagram is very helpful for readers in understanding your search and screening process. I have a few clarifications:
o It is not clearly described in the methods section how the abstracts were originally identified in the IADR Abstract Archives. From the flow diagram, it looks like a general search for RCTs was applied, but this was not described in the methods. In the “records identified through searching” box, what was the total number of abstracts that you started with, and how many did you identify through a search? I see n=2018, but I’m not sure what that is referring to especially because it says in the “Records after duplicates removed” box that you had 6619 to screen.
o In the “records excluded” box, it is best practice to provide how many abstracts were excluded based on each reason.
Figure 2
• Based on the text, I have an idea what this figure is depicting, but it is not clear on its own. Consider if there is another way to visualize this data or at least clearly label the figure components.

Raw data supplied
• Thank you for providing the raw data. I suggest including more metadata to describe your dataset so that readers can understand your data and make use of it. Here is some guidance on writing a readme style metadata document: https://data.research.cornell.edu/content/readme.

Experimental design

Research question well defined, relevant & meaningful. It is stated how the research fills an identified knowledge gap.
• The authors demonstrate how their research fills a research gap and have a clear study objective on lines 74-76. 1. As mentioned previously, it could use additional sentence clearly indicating research aims. They may also want to clarify in their objective statement that they are investigating prosthodontics RCTs presented at the IADR. The IADR is mentioned in the first paragraph but not again in the introduction. Why prosthodontics was selected was also not mentioned.

Rigorous investigation performed to a high technical & ethical standard.
• The authors were thorough and rigorous in their methods. Their process for retrieving and matching full-text articles with abstracts is comprehensive and well thought out. The supplemental files show that the authors were detailed and thorough in the matching process, and I appreciate that they took the time to document the matched abstracts and provide that information to readers.

Methods described with sufficient detail & information to replicate.
• As mentioned in the figures section, the ‘selection of conference abstracts’ subsection needs some clarification. The authors provide inclusion and exclusion criteria for the abstracts, but do not describe how the identified the abstracts to begin with. Was it a search, as suggested by Figure 1, or did they screen the abstracts? As there were over 10,000 abstracts this would have been a huge undertaking and could be described in more detail.
• In the results section lines 166 to 173, the authors describe how they calculated inconsistences between conference and published abstracts. This type of information should be in the methods, and the results of the inconsistency data should be in the results.
• The exclusion criteria in the selection of conference abstracts subsection should match the exclusion criteria in the flow diagram (Figure 1).

Validity of the findings

• In the discussion and conclusion sections, the authors clearly link to their research question by discussing what items had the highest and lowest consistency, and the predictors of inconsistency between abstracts. They also offer explanations for the results, based on the literature and some speculation.
• Another limitation that was not mentioned is that authors only compared published article abstracts and conference abstracts. It is possible that the discrepancies and reasonable explanations for them were described in the manuscript, but not in the abstract.
• The first sentence of the discussion section (179-180) needs a citation. This article could work, but I encourage you to look for others: https://health-policy-systems.biomedcentral.com/articles/10.1186/1478-4505-10-26
• Since you used CONSORT-A to rate the abstracts, you could re-iterate that authors closely follow CONSORT-A and other relevant reporting guidelines when writing abstracts and manuscripts.

Additional comments

This is a well-designed study investigating the discrepancies between conference and journal abstracts. While this study is limited to prosthodontics abstracts, its results are a good reminder to researchers across disciplines to be cautious when applying findings heard at conferences, and to be more transparent and thorough when reporting their research in manuscripts. The main weakness is that the authors did not evaluate the full-text of the published article, which would have provided more context for inconsistencies between conference and published versions. The authors could improve their work by clarifying certain steps in their methods and having a colleague proficient in English improve some of the phrasing.

---

## Round 0.2 · accepted · Accept

The authors have addressed the reviewers' comments.

·

Basic reporting

no comments, the authors have addressed concerns

Experimental design

no comments, the authors have addressed concerns

Validity of the findings

no comments, the authors have addressed concerns